# The Tumor-Associated Calcium Signal Transducer 2 (TACSTD2) oncogene is upregulated in cystic epithelial cells revealing a potential new target for polycystic kidney disease

Abigail O. Smith[1,2☉], William Tyler Frantz[1,2☉], Kenley M. Preval[1,2], Yvonne J. K. Edwards[1], Craig J. Ceol[1], Julie A. Jonassen[3], Gregory J. Pazour[1]*

**1** Program in Molecular Medicine, University of Massachusetts Chan Medical School, Worcester, Massachusetts, United States of America, **2** Morningside Graduate School of Biological Sciences, University of Massachusetts Chan Medical School, Worcester, Massachusetts, United States of America, **3** Department of Microbiology and Physiology Systems, University of Massachusetts Chan Medical School, Worcester, Massachusetts, United States of America

☉ These authors contributed equally to this work.
* gregory.pazour@umassmed.edu

**Data Availability Statement:** The data discussed in this publication have been deposited in NCBI's

## Abstract

Polycystic kidney disease (PKD) is an important cause of kidney failure, but treatment options are limited. While later stages of the disease have been extensively studied, mechanisms driving the initial conversion of kidney tubules into cysts are not understood. To identify genes with the potential to promote cyst initiation, we deleted polycystin-2 (*Pkd2*) in mice and surveyed transcriptional changes before and immediately after cysts developed. We identified 74 genes which we term cyst initiation candidates (CICs). To identify conserved changes with relevance to human disease we compared these murine CICs to single cell transcriptomic data derived from patients with PKD and from healthy controls. *Tumor-associated calcium signal transducer 2* (*Tacstd2*) stood out as an epithelial-expressed gene with elevated levels early in cystic transformation that further increased with disease progression. Human tissue biopsies and organoids show that TACSTD2 protein is low in normal kidney cells but is elevated in cyst lining cells, making it an excellent candidate for mechanistic exploration of its role in cyst initiation. While TACSTD2 has not been studied in PKD, it has been studied in cancer where it is highly expressed in solid tumors while showing minimal expression in normal tissue. This property is being exploited by antibody drug conjugates that target TACSTD2 for the delivery of cytotoxic drugs. Our finding that *Tacstd2/TACSTD2* is prevalent in cysts, but not normal tissue, suggests that it should be explored as a candidate for drug development in PKD. More immediately, our work suggests that PKD patients undergoing TACSTD2-directed treatment for breast and urothelial cancer should be monitored for kidney effects.

Gene Expression Omnibus and are accessible through GEO Series accession numbers: GSE222610 (https://www.ncbi.nlm.nih.gov/geo/query/acc.cgi?acc=GSE222610) GSE220322 (https://www.ncbi.nlm.nih.gov/geo/query/acc.cgi?acc=GSE220322).

**Funding:** Work was supported by NIH DK103632 (GJP) and GM060992 (GJP) including salary support for AOS, KMP, and GJP. The funders had no role in study design, data collection and analysis, decision to publish, or preparation of the manuscript.

**Competing interests:** I have read the journal's policy and the authors of this manuscript have the following competing interests: A provisional patent has been filed: US Application No.: 63/589,316 THERAPY FOR POLYCYSTIC KIDNEY DISEASE

## Author summary

Polycystic kidney disease (PKD) causes kidney failure through the growth of large, fluid-filled cysts. The most common type is inherited in a dominant manner with pathogenic variants in a single allele causing disease. Although the genes responsible for PKD have been known for years, early cellular and genetic changes in cyst formation are not well understood. In our study, we sequenced the kidney transcriptomes of mice with PKD both before and after cysts formed and compared them to normal kidneys of the same age. We identified a small number of genes with altered expression before visible cyst formation including Tumor associated calcium signal transducer 2 (Tacstd2). Immunostaining confirmed Tacstd2 is elevated in pre-cystic mouse kidneys and that it is abundant in cysts from human biopsies and human kidney organoids. Tacstd2 has not been studied in relation to kidney disease but has been the subject of extensive research in cancer where it has been found have high expression in many solid tumors with little expression in normal tissues. This observation is being exploited to deliver cytotoxic drugs specifically to abnormal cells, suggesting that similar methods could be developed for PKD treatment.

## Introduction

Autosomal Dominant Polycystic Kidney Disease (ADPKD) is an inherited, progressive disease in which large, fluid-filled cysts grow in both kidneys, ultimately destroying organ function. It is relatively common, affecting approximately 1 in 1,000 people [1,2,3]. ADPKD typically results from inactivating mutations in genes encoding either of two transmembrane proteins, polycystin-1 (PKD1 in humans, Pkd1 in mice) or polycystin-2 (PKD2/Pkd2). The precise function of these channel-like proteins is unknown, but they are thought to regulate proliferation and differentiation to maintain tubular architecture. Numerous pathways downstream of the polycystins have been identified including cAMP signaling, which is the basis of Tolvaptan, the only approved FDA-approved drug therapy for ADPKD [4,5]. Nonetheless, most patients require dialysis or kidney transplant by late adulthood [6].

While ADPKD is categorized as an adult disease [7], cyst formation begins *in utero* and continues throughout an individual's lifespan. Adolescents may experience early symptoms of hypertension and pain, and symptom prevalence increases with age and cyst burden despite preserved kidney function [8,9]. Ultimately cyst burden overwhelms kidney function, leading to organ failure.

While the later stages of the disease have been extensively studied, there has been limited research into the early stages of ADPKD, including the mechanism of cyst initiation. Identifying drivers of cyst formation could provide additional therapeutic targets. In this work we used RNA sequencing to detect the earliest transcriptional changes after *Pkd2* deletion in mouse. We identified 74 differentially expressed cyst initiation candidates (CICs). Cross-referencing CICs with published data from pre-cystic mouse models and single-cell RNA sequence from murine and human kidneys pointed to *Tacstd2/TACSTD2*, also known as *Trop2* as a strong candidate for further investigation. While unstudied in the context of ADPKD, *TACSTD2* is highly expressed in metastatic breast cancer and other epithelial tumors. It is targeted by Sacituzumab govitecan (Trodelvy), an antibody-drug conjugate as a treatment for aggressive cancer [10]. Tacstd2 antibodies strongly label small cysts in both human and mouse kidneys with minimal label in non-cystic tubules indicating that this protein should be explored as a therapeutic target.

## Results

### Identifying early transcriptional changes after *Pkd2* deletion

To distinguish the earliest transcriptional changes downstream of the loss of *Pkd2* function we utilized a conditionally inducible mouse model [11] (Fig 1). After intraperitoneal tamoxifen injection at postnatal day 2 (P2), control (*Cagg-Cre^ER*, *Pkd2^flox/+*  or *Pkd2^flox/null*, [no cre]) and experimental (*Cagg-Cre^ER*, *Pkd2^flox/null*) kidneys were studied at P6 and P10 (Fig 1B–1C), and P14-15 (Fig 1C). The two-kidney to bodyweight ratio (2K,BW), which is a proxy for cystic burden, was not significantly different between experimental and control animals at P6, but by P10 2K:BW was significantly elevated in the experimental animals and was further elevated in P14-15 experimental animals (Fig 1C). Consistent with the 2K:BW, H&E staining revealed no gross differences in kidney architecture in P6 experimental kidneys (Fig 1B), but cysts were evident in the P10 experimental kidneys (Fig 1B). At P10, experimental collecting ducts (marked by Aquaporin-2) were dilated or cystic whereas collecting ducts appeared normal in P6 experimental kidneys (Fig 1D).

To identify potential drivers of cyst formation, RNA was isolated from P6 (n = 28) and P10 (n = 20) experimental and control kidneys with sex parity (Table 1). mRNA and small RNA libraries were generated, sequenced, and analyzed by standard pipelines to identify transcriptional changes (Fig 1A). For mRNA, differentially expressed genes (DEGs) were defined as having a false discovery rate (FDR) of less than 0.05. At P6 there were 91 DEGs (59 increased expression and 32 decreased expression) (Fig 2A and S1 Table). At P10 there were 5,309 DEGs (2,645 increased expression and 2,664 decreased expression) (Fig 2B and S2 Table). *Pkd2* mRNA was significantly reduced at both time points, validating our inducible Cre system. The large difference between the number of DEGs at P10 vs P6 likely corresponds to the observed rapid cyst development within this time frame.

The scope of this work was to identify and characterize strong candidates for future mechanistic and possibly therapeutic studies with regards to cyst initiation. We predicted that the P10 timepoint with gross cystic changes and many differentially expressed genes would identify gene expression patterns related to downstream cyst-related changes rather than cyst initiation. In contrast, P6 differences, although fewer and of smaller magnitude than P10, are more likely to represent cyst initiation. We theorized that persistence from P6 to P10 during cyst development strengthens the possibility that the genes participate in cyst initiation. We identified 74 genes in the intersection between P6 and P10 DEGs. We defined this list as cyst initiation candidates (CICs) (Fig 2C and 2D). The vast majority of CICs are regulated in the same direction at P6 and P10, suggesting that the changes detected at P6 increase with cyst development (Fig 2D and 2E).

We used functional enrichment analysis to characterize the differentially expressed genes in each category: unique to P6, common to P6 and P10 (CICs) and unique to P10 (Figs 2C and S2A). We used g:Profiler g:Gost [12] to find enrichment of a ranked gene list in multiple databases simultaneously. Our analysis included Gene Ontology (GO) molecular function–(MF), biological process (BP) and cellular component (CC) as well as Kyoto Encyclopedia of Genes and Genomes (KEGG) and Reactome (REAC) and WikiPathways (WP). The CICs were enriched in functions including integrin binding involved in cell-matrix adhesion (*Svep1*, *Itga9*), cell motility (*Svep1*, *Arpin*, *Sema3c*, *Srpx2*, *Itga9*, *Akap12*, **Tacstd2**, *Bcl11b*, *Mgat5*, *Cfap20*, *Ccdc25*, *Vil1*, *Vstm4*, *Ccn4*, *Ntn4*, *Dnah5*), and extracellular matrix (*Fbln5*, *Svep1*, *Sema3c*, *Srpx2*, *Plxdc2*, *Lad1*). The 17 genes unique to P6 were enriched in GO terms related to extracellular matrix formation (*Fbn1*, *Col14a1*, *Col6a3*, *Sparcl1*) and lipid transport (*Spns2*, *Plscr4*, *Mttp*). The top 1000 DEGs unique to P10 were analyzed. As expected, these corresponded to broad GO terms and pathways involved in cellular proliferation including protein

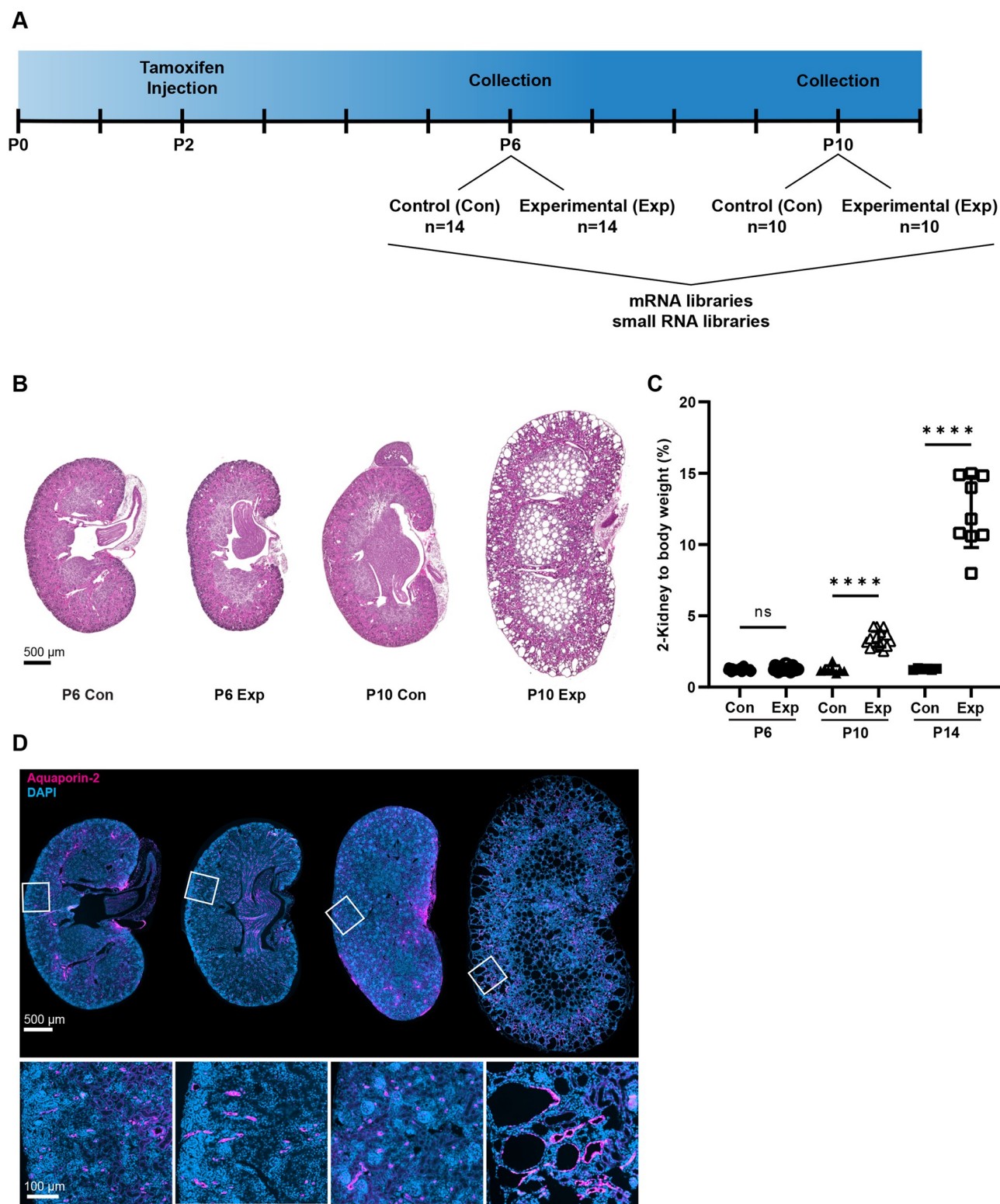

**Fig 1. *Pkd2* deletion at postnatal day 2 (P2) induces rapid cyst growth.** Mouse genotypes: Con (*Cagg-Cre^ER*, *Pkd2^{flox/+}* or Cre-negative, *Pkd2^{flox/null}*), Exp (*Cagg-Cre^ER*, *Pkd2^{flox/null}*). (A) Tamoxifen induction occurred at P2 followed by kidney collection for RNA sequencing at P6 and P10. Parallel mRNA and small RNA libraries were prepared. (B) H&E staining show the extent of disrupted organ architecture. Images captured by Zeiss Axio Scan. Z1 with 20X objective. Scale bar is 500 microns and applies to all images in the panel. (C) Cystic burden was quantified using the ratio of 2-kidney weight / body weight x 100%. N is 28 (P6 Con), 27 (P6 Exp), 8 (P10 Con), 16 (P10 Exp), 6 (P14 Con), 9 (P14 Exp). ****, p < 0.0001; ns, not significant

by one-way ANOVA followed by Tukey multiple comparison test with multiplicity-adjusted p-values. Error bars indicate SD. (D) Kidney sections were probed for collecting duct marker Aquaporin-2 (red). Nuclei were marked by DAPI. Images captured by Zeiss Axio Scan.Z1 with 20X objective. Insets from indicated regions show detail of collecting duct at higher power. Scale bar is 500 microns for full size images, 100 microns for enlargements.

binding, cell cycle, cytoskeleton, and DNA replication. Results suggest that Pkd2 loss leads to early changes in cell motility programs, involving extracellular matrix. However, most CICs were not enriched in established functions, prompting us to seek alternative methods of characterizing these genes.

There were no differentially expressed miRNAs in the P6 dataset that reached significance (FDR < 0.05). At this early pre-cystic time point any differences in miRNA expression are likely subtle and below the level of detection by bulk sequencing. Nevertheless, we found 39 stem-loop and mature miRNAs with FDR < 0.05 at P10 (S1A Fig). 24 of the 39 genes were previously cited in PKD literature (S1B Fig), further validating our ability to detect relevant changes in small RNA expression between experimental and control kidneys. The most significantly elevated miRNA, mir-551b, is implicated in abnormal cell differentiation and proliferation in gastric cancer [13] and ovarian cancer [14]. Mir-551b may warrant further study in relation to PKD pathogenesis. However, the remainder of this study focuses on mRNAs with differential expression in pre-cystic kidneys as novel candidates for cyst initiation.

## Exploring cyst initiation candidates in published RNAseq datasets

To filter candidate genes by predicted probability of participation in early cystic transformation, we leveraged existing data from the field. We reasoned that strong cyst initiators would be consistent in alternative mouse models of early PKD. Therefore, we compared our data to models differing in Cre expression, target gene (*Pkd2* vs. *Pkd1*), age and method of induction, age at sacrifice, cyst burden at time of collection, and number of mice used for analysis (Table 2). Woo *et al.* [15] looked at disease progression in young mice at P1, P3 and P7 where *Pkd1* or *Pkd2* had been deleted during development of the collecting duct with *HoxB7-Cre*. This study observed few significant transcriptomic changes at the earliest time points after polycystin inactivation and the changes observed did not correlate strongly with our DEGs (S2B and S2C Fig). However, two other studies looking at gene expression in pre-cystic adult mouse models correlated significantly with our data at P6 and P10 (Figs 2E, S2B and S2C). Kunnen *et al.* [16] deleted *Pkd1* in adult mouse kidneys via *Ksp-Cre* [16] and Zhang *et al.* [17] deleted *Pkd2* in collecting ducts and distal tubules of mature adult mice with doxycycline-inducible *Pax8*[rtTA], *TetO-Cre* [17]. Remarkably, we observed overall agreement in pre-cystic kidneys even between *Pkd2* and *Pkd1* models and juvenile and mature mice.

## Cyst initiation candidates in human disease

To focus on the genes most relevant to human disease, we identified CICs with conserved expression patterns by reanalyzing Muto *et al.'s* [18] single cell data derived from healthy and

**Table 1. Experimental design.**

| Timepoints | PKD2 Control (Con) *Cagg-Cre*[ER],*Pkd2*[+/flox] (n = 20) or *Pkd2*[null/flox], [no Cre] (n = 4) | PKD2 Deletion (Exp) *Cagg-Cre*[ER], *Pkd2*[null/flox] (n = 24) |
|---|---|---|
| P6 (n = 28) | Male (n = 7) Female (n = 7) Total (n = 14) | Male (n = 7) Female (n = 7) Total (n = 14) |
| P10 (n = 20) | Male (n = 3) Female (n = 7) Total (n = 10) | Male (n = 5) Female (n = 5) Total (n = 10) |

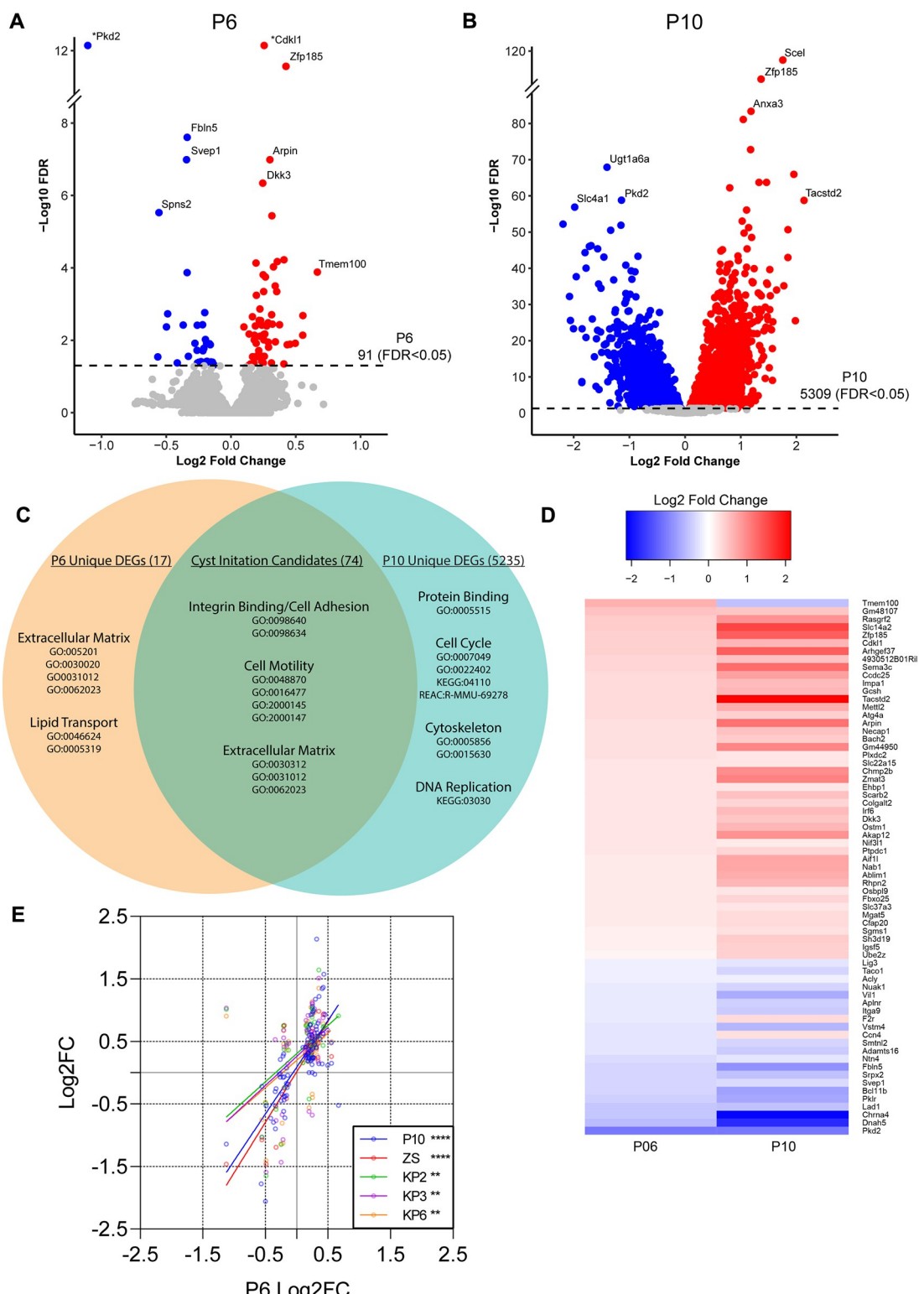

**Fig 2. Identifying cyst initiation candidate genes through mRNA sequencing *Pkd2* mouse kidneys at pre-cystic and early cystic time points.** (A) Volcano plot showing pre-cystic (P6) differentially expressed genes (DEGs; DESeq2). Colored dots indicate FDR <0.05, red indicates increased and blue decreased in Experimental vs Control. Plot was cropped for better visualization. (*) Data point moved for better visualization: Pkd2 [Log2(FC) = -1.13, -Log10(FDR) = 93.97], Cdkl1 [Log2(FC) = 0.42, -Log10(FDR) = 26.30]. (B) Volcano plot showing early cystic (P10) DEGs (DESeq2). Colored dots indicate FDR <0.05, red

indicates increased expression and blue decreased expression in Experimental vs Control. (C) The intersection between P6 DEGs (91) and P10 DEGs (5,309) includes 74 genes which we defined as cyst initiation candidates (CICs). We performed functional enrichment analysis to identify the top cellular functions associated with each list of DEGs: unique to P6, common to P6 and P10 (CICs) and unique to P10. Genes were ranked by padj values and P6 padj values were used for ranking CICs. False Discovery Rate (FDR) method was used to control for false positives with a threshold of 0.01. (D) Heatmap showing the relative expression of all CICs at P6(left) and P10 (right). (E) Scatter plot showing correlation of CICs with published data sets. P6 DEG expression (Log2FC) represented along x-axis vs. P10 and published datasets on y-axis (Log2FC). Datasets include ZS (*Pkd2*) [17] and KP2/3/6 (*Pkd1*; 2 weeks, 3 weeks and 6 weeks post-induction respectively) [16]. Details of each study can be found in Table 2. There is significant correlation between P6 and each group by simple linear regression analysis (two-tailed p-values: P10, $p<0.0001$; ZS, $p<0.0001$; KP2, $p = 0.0014$; KP3, $p = 0.0016$; KP6, $p = 0.0012$).

ADPKD human kidneys. The Muto *et al.* [18] data was re-calculated and cell types assigned according to consensus markers from Chen *et al.* [19] (Fig 3A). To compare transcriptional differences present in human disease, we first pseudo bulked Muto et al.'s [18] data to identify differentially expressed genes and then compared these genes to our list of 74 CICs (Fig 3B). We found 30 DEGs in common between the human data and our list of CICs. The majority agreed in direction between human disease and our early cystic murine model. Others (such as *Cdkl1*) demonstrated differential expressions in directions counter to our observations (Fig 3B). We remain interested in the genes with discrepant expression, as currently available human data necessarily derives from later disease stages than mouse studies, and thus may not fully constitute the milieu of cyst initiation.

Factors that drive initial cyst growth are likely to be expressed in epithelial cells as these are sites of the first pathology in cystic disease [20]. Single cell sequence data from mouse kidney has more clearly defined cell types than equivalent human data allowing for better assignment of genes to cell types. Thus to identify the particular cell types where our CICs are expressed, we reanalyzed published single cell transcriptomes from healthy mouse kidneys [21] to construct a uniform manifold projection (Fig 3C) where the identities of cell clusters were assigned using consensus marker genes from Chen *et al.* [19] (S3B Fig). The clusters broadly agreed with original cell type assignments by Ransick *et al.* [21] (S3A–S3C Fig). The three main expression patterns we found include genes restricted to epithelial cells (genes such as *Tacstd2*, *Cdkl1*, *Akap12*, and *Aif1L*), those with broad expression in many cells (*Osbpl9*), and those expressed in stromal cells (Endothelium: *F2R*; T-cell: *Bcl11B*) (Fig 3D).

**Table 2. Early and pre-cystic mouse models with RNAseq data.**

| Publication | Cre | Target Gene | Induction conditions | Age at sacrifice | Degree of Cysts |
|---|---|---|---|---|---|
| This Work | *Cagg-Cre^ER*–whole body | *Pkd2* | Tamoxifen IP at P2 | P6 | Pre |
| " | *Cagg-Cre^ER*–whole body | *Pkd2* | Tamoxifen IP at P2 | P10 | Early |
| Kunnen *et al.* [16] | *KspCad-Cre^ERT2* –kidney specific | *Pkd1* | Tamoxifen in food 5mg/day x 3 days at 13–14 weeks of age | 13/14 + 2 weeks | Pre |
| " | *KspCad-Cre^ERT2* –kidney specific | *Pkd1* | Tamoxifen in food 5mg/day x 3 days at 13–14 weeks of age | 13/14 + 3 weeks | Pre |
| " | *KspCad-Cre^ERT2* –kidney specific | *Pkd1* | Tamoxifen in food 5mg/day x 3 days at 13–14 weeks of age | 13/14 + 6 weeks | Pre |
| Zhang *et al.* [17] | Pax8^rtTA,TetO-Cre–kidney selective, whole nephron | *Pkd2* | Doxycycline in drinking water + sucrose 2mg/mL for 2 weeks starting at P28-P42 | P70 | Pre |
| Woo *et al.* [15] | *HoxB7-Cre*–collecting duct | *Pkd2* | *HoxB7-Cre* becomes active E9.5-E12.5 | P1 | Pre |
| " | *HoxB7-Cre*–collecting duct | *Pkd2* | *HoxB7-Cre* becomes active E9.5-E12.5 | P3 | Early |
| " | *HoxB7-Cre*–collecting duct | *Pkd2* | *HoxB7-Cre* becomes active E9.5-E12.5 | P7 | Mid |
| " | *HoxB7-Cre*–collecting duct | *Pkd1* | *HoxB7-Cre* becomes active E9.5-E12.5 | P1 | Pre |
| " | *HoxB7-Cre*–collecting duct | *Pkd1* | *HoxB7-Cre* becomes active E9.5-E12.5 | P3 | Early |
| " | *HoxB7-Cre*–collecting duct | *Pkd1* | *HoxB7-Cre* becomes active E9.5-E12.5 | P7 | Mid |

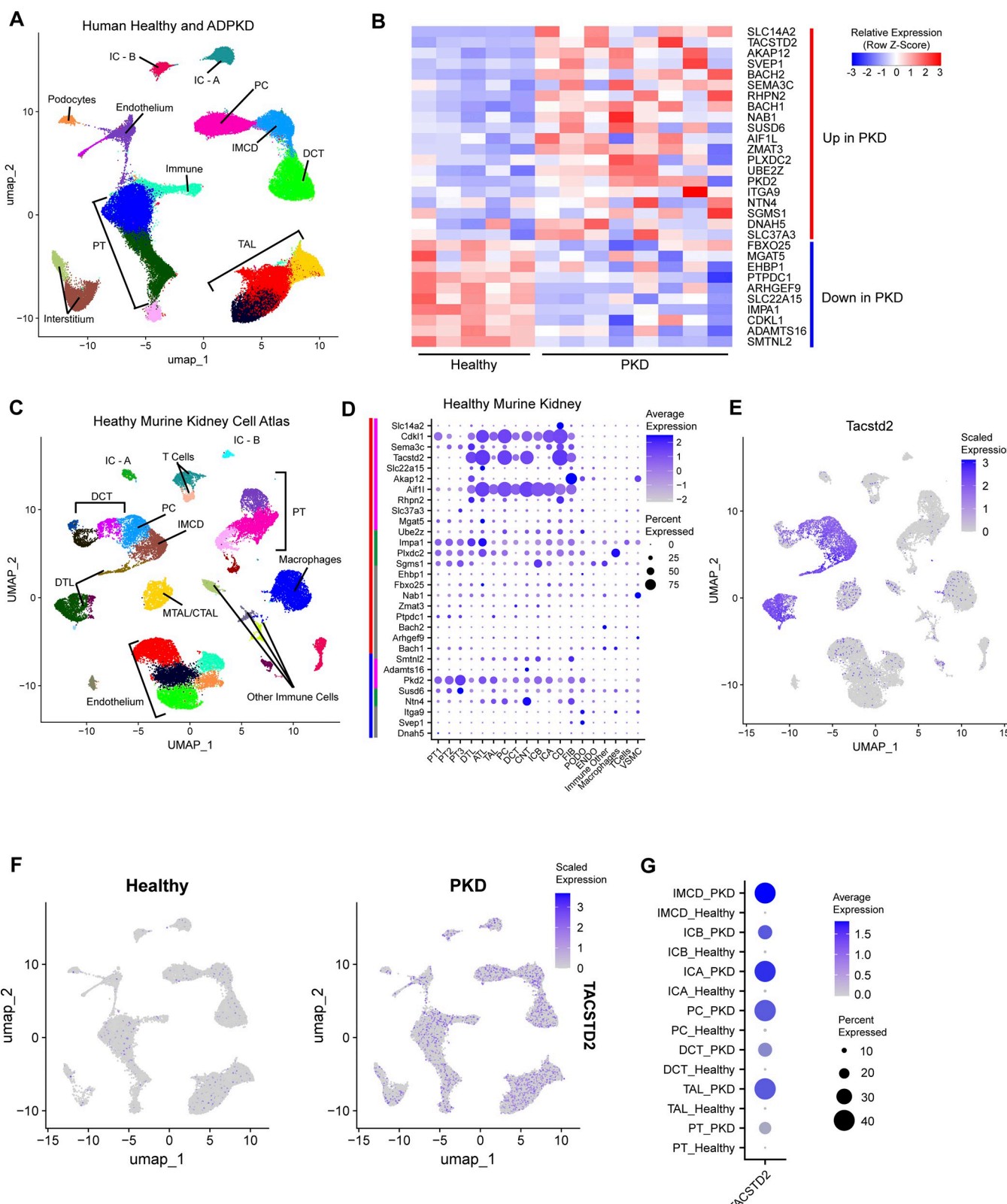

**Fig 3. Leveraging multispecies single cell RNAseq to identify tubular cystogenic targets.** (A) UMAP visualization of human kidney cells from healthy and ADPKD tissue [18]. DCT: Distal Convoluted Tubule, PT: Proximal Tubule, IC-A: Intercalated Cell–A, IC-B: Intercalated Cell–B, PC: Principal Cell, IMCD: Inner Medullary Collecting Duct, TAL: Thick Ascending Limb of Henle. (B) Pseudobulk heatmap showing expression of candidate genes (based on murine

data in Fig 2) in healthy (n = 5) vs ADPKD (n = 8) human kidney tissue [18]. The candidates shown are differentially expressed using the Wilcox sum in "FindMarkers" and ordered by log2FC. FDR < 0.01. Red bar signifies up in human ADPKD kidney vs healthy control kidney. Blue bar signifies down in human ADPKD vs healthy control kidney. Row-normalized expression. (C) UMAP visualization of murine kidney cells from Ransick et al.'s Kidney Cell Atlas [21]. DCT: Distal Convoluted Tubule, PT: Proximal Tubule, IC-A: Intercalated Cell–A, IC-B: Intercalated Cell–B, PC: Principal Cell, IMCD: Inner Medullary Collecting Duct, DTL: Descending Thin Limb of Henle, ATL: Ascending Thin Limb of Henle, MTAL/CTAL: Medullary Thick Ascending Limb of Henle/ Cortical Thick Ascending Limb of Henle. (D) DotPlot visualizing the localization of conserved murine/human differentially expressed CICs to cell types found in the murine kidney cell atlas. Dot radius represents the percentage of cells in cluster expressing the gene. Opacity scale indicates average gene expression level within expressing cells. Row normalized. Main bars (far left) represent increased expression (red) or decreased expression (blue) in our data; sub bars (left) represent epithelial restricted expression (magenta), broad expression (green), and stromal/indeterminate (gray). Black arrow points to *Tacstd2*, an epithelial cyst initiating candidate. (E) Feature plot highlighting the expression profile for *Tacstd2* in healthy mouse kidney. (F) Feature plot highlighting the diffusely increased expression profile for *TACSTD2* in ADPKD vs healthy human kidneys. (G) DotPlot comparing *TACSTD2* expression in human healthy and ADPKD kidneys as seen in C by cell type labeled in A [18].

When inspecting CICs expressed in epithelial cells and showing differential expression in human PKD, the tumor associated calcium signal transducer *Tacstd2* stood out. This gene, whose expression was restricted to epithelial cells of the distal tubule (Fig 3D and 3E) was one of the most dysregulated genes in human PKD (FDR <1E$^{-10}$) (Fig 3B and S3 Table). In our data, *Tacstd2* was dysregulated in P6 pre-cystic kidneys, P10 early cystic kidneys, and in early time points of Woo *et al.*'s [15] *Pkd1* AND *Pkd2* models (S2B Fig). In healthy human kidneys most cells show low to undetectable expression, but the expression goes up considerably in the distal parts of the uriniferous tubule in PKD tissues (Fig 3F and 3G).

## Characterizing Tacstd2 as a cyst initiation candidate

To characterize the epithelial expression of *Tacstd2* predicted by single-cell RNAseq, we immuno-stained kidney tissues collected from the same cohort of animals that were used for RNA isolation in this study (Fig 4A, 4B and 4D). In addition, to see how *Tacstd2* responds in more highly cystic kidneys, we examined P21 tissue collected from another study where *Pkd2* was deleted in the early post-natal period by tamoxifen treatment of mothers nursing *Rosa26-Cre$^{ERT2}$*, *Pkd2$^{flox/flox}$* pups [22]. We find that Tacstd2 does not stain proximal tubules in either control or *Pkd2* mutant tissue. The loop of Henle shows only minimal Tacstd2 label in the pre cystic tissue, but loop of Henle-derived cysts show Tacstd2 label (Fig 4D). Tacstd2 labels collecting ducts in both control and experimental kidneys (Fig 4B and 4D). While Tacstd2 is found in control collecting ducts, the signal is higher in the experimental collecting ducts at all time points (Fig 4B and 4C). When normalized by DAPI signal to represent the total number of cells, Tacstd2 is significantly increased in experimental compared to control at P10 and P21. At P6 the difference did not reach significance (Fig 4C). All images used for quantification are included in S4 Fig.

The elevated Tacstd2 expression in our *Pkd2* mutant mouse kidneys prompted us to ask whether TACSTD2 expression was also elevated in human ADPKD samples. To do this, we obtained sections of adult male non-cystic and cystic kidneys. The genotypes are unknown, but the early disease development in these patients is suggestive of *PKD1* mutations. Similar to what we observed in our murine models, TACSTD2 was only faintly detected in control tissues but was strongly observed in the cyst-lining epithelial tissue from cystic patients and overall levels of TACSTD2 signal were higher in the cystic kidneys (Fig 5A and 5B).

Since we were unable to determine the genotypes of the patient samples, we wanted to learn whether TACSTD2 is elevated by polycystin loss in human kidney organoids with defined genotypes. To do this, we used control, *PKD1$^{-/-}$*, and *PKD2$^{-/-}$* IPS cells generated by the PKD Research Resource Consortium to generate organoids using a slightly modified Ruiter *et al.* protocol [23]. This protocol produced robust cysts from both mutant lines although the *PKD1$^{-/-}$* derived organoids were more extensively cystic (Fig 5C). Control organoids showed

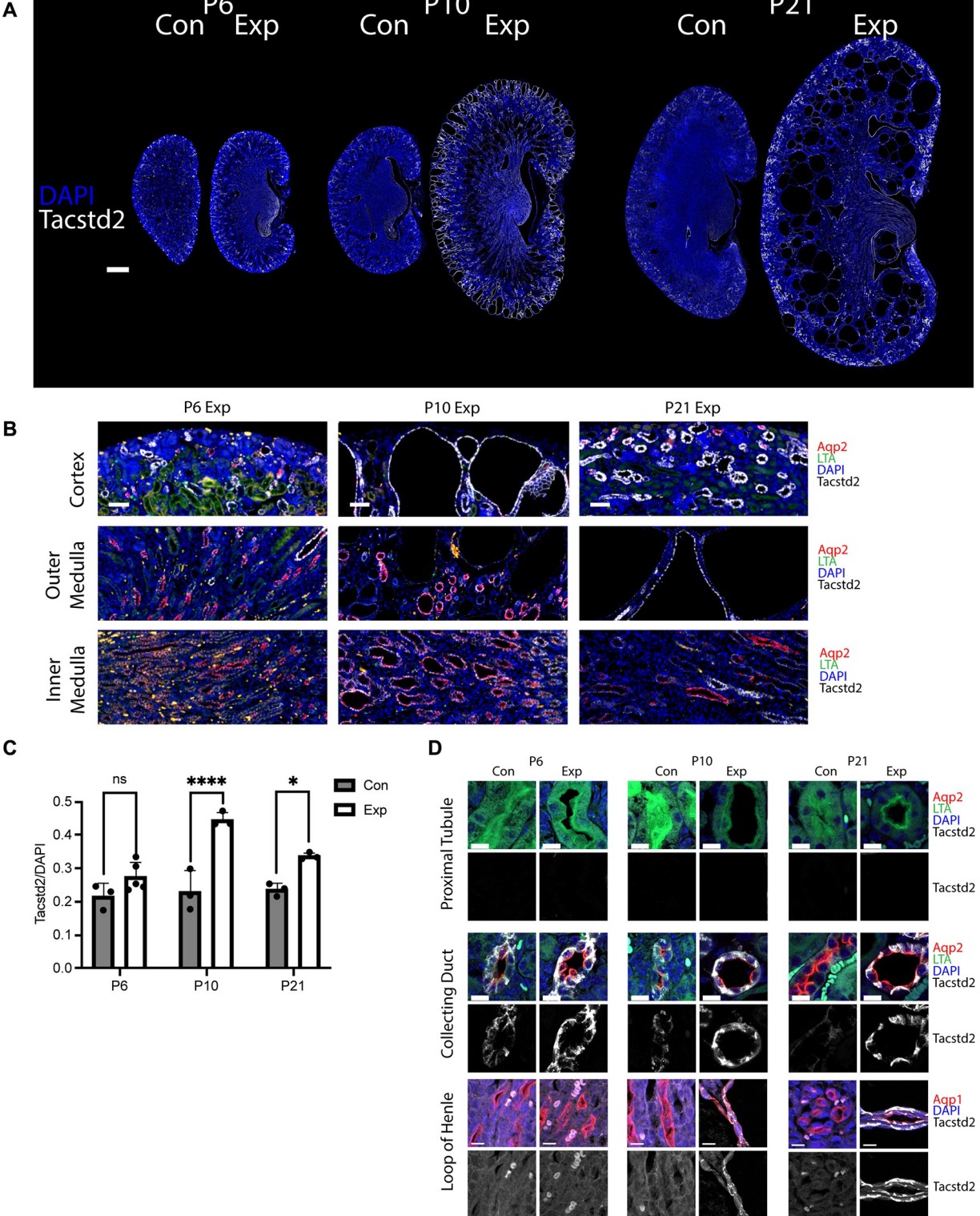

**Fig 4. Characterizing *Tacstd2* expression in mouse kidney epithelium.** (A) Mouse kidney sections were probed for Tacstd2 (fuchsia) and DNA (DAPI, blue). Slide scans were captured by Zeiss Axio Scan.Z1 with 20X objective. Scale bars, 500 microns. (B) Tissue was stained for the collecting duct marker aquaporin2 (Aqp2, red), proximal tubule marker *Lotus tetragonolobus* Agglutinin (LTA, green), nuclei (DAPI, blue) and Tacstd2 (white). Tissues scans were captured by Zeiss Axio Scan.Z1 with 20X objective. Scale bars 50 microns. (C) Mean intensity of Tacstd2 fluorescence was measured for Control (Con) and Experimental (Exp) mouse kidneys from three age groups: P6, P10 and P21. Normalized by

mean intensity of DAPI. P10, P21 and P6 WT (n = 3); P6 KO (n = 5). Mixed effects analysis with multiple comparisons was performed. Factors included age (p = 0.0019) and genotype (p<0.0001). Non-significant, ns; *, p<0.05; ****, p<0.0001. (D) Tissue was stained for the collecting duct marker aquaporin2 (Aqp2, red), proximal tubule marker *Lotus tetragonolobus* Agglutinin (LTA, green), nuclei (DAPI, blue) and Tacstd2 (white) in the proximal tubule and collecting duct groups. In the loop of Henle group, tissue was stained with aquaporin1 (Aqp1, red), nuclei (DAPI, blue) and Tacstd2 (white). Bottom row of each pair shows the Tacstd2 channel in grey scale. Aquaporin1 also stains proximal tubules and vasa recta but loop of Henle is distinguished by the lack of a brush border and the absence of red blood cells. Note that red blood cells appear lime green in the proximal tubule and collecting duct groups and light pink in the loop of Henle group due to autofluorescence. Tissues were imaged using AiryScan LSM 900 with 63X objective. Maximum intensity projections of a 5-micron section imaged at 0.3 micron intervals. Scale bars, 10 microns.

little staining with TACSTD2 antibodies while *PKD1*[-/-] and *PKD2*[-/-] organoids showed robust TACSTD2 label of the tubules and cysts. The very large cysts that develop when *PKD1*[-/-] is mutated were strongly stained (Fig 5D and 5E). TACSTD2 stained the whole cell with highest concentration at the cell surface (Fig 5D). From this analysis, we conclude that TACSTD2 levels increase with loss of either *PKD1* or *PKD2*.

## Discussion

Improving ADPKD treatment and outcomes depends on understanding genetic changes that promote cyst development. Focusing on meaningful changes in kidney gene expression that precede cyst formation has been challenging due to low amplitude gene expression within low abundance cell types in heterogeneous tissues. Previous investigations [15] found few genes differentially expressed in pre-cystic time points but found many differentially expressed genes later disease stages. However, the genes elevated in grossly cystic tissue generally participate in broad proliferative pathways, and do not inform the mechanisms of cyst initiation.

With these previous studies and pitfalls in mind, we set out to identify potential cyst drivers by controlling the timing of *Pkd2* loss. To find low amplitude changes present before gross cyst formation, we utilized a high number of animals with equal sex parity. We then cross referenced our data to previous studies to find genes with consistent changes. We further cross referenced our differentially expressed genes against human disease and murine single cell atlases to focus on conserved cystogenic changes in epithelial cells. Through our multispecies, multiomic approach, we identified 74 cyst initiation candidates including tumor-associated calcium signal transducer 2 (*Tacstd2*).

This study faces several limitations that should be considered when interpreting the results. A significant challenge is the scarcity of human ADPKD kidney samples in the pre-cystic or early cystic stages, which constrains our ability to fully understand early disease mechanisms. Additionally, while the chosen mouse model offers valuable insights, it has limitations in fully mimicking human disease. We explored alternative models, such as those with adult onset, kidney-specific Cre, and Pkd1 deletion, but the practicalities of parallel sequencing across these options were prohibitive. We opted for a model with systemic *Pkd2* deletion and cyst initiation in young, developing kidneys. While ADPKD is normally thought of as an adult onset disease, cysts initiate *in utero* and the extent of early cysts is likely a driver of disease severity [24,25,26,27]. Despite its limitations, this model offers valuable insights into human disease. To bolster our findings, we complemented our data with high-quality published research. Importantly, this study does not establish whether Tacstd2 is necessary or sufficient for cyst initiation, a question that will be addressed in future research. Nonetheless, even if Tacstd2 is an early consequence rather than a cause of cyst formation, its role as a marker of cystic epithelium could prove beneficial for targeting anti-cyst therapies to affected tissues.

We observed that *Tacstd2* is robustly elevated in Woo *et al.* [15], but not in Kunnen *et al.* [16] or Zhang *et al.* [17]. A major difference between the studies is the age at which polycystin

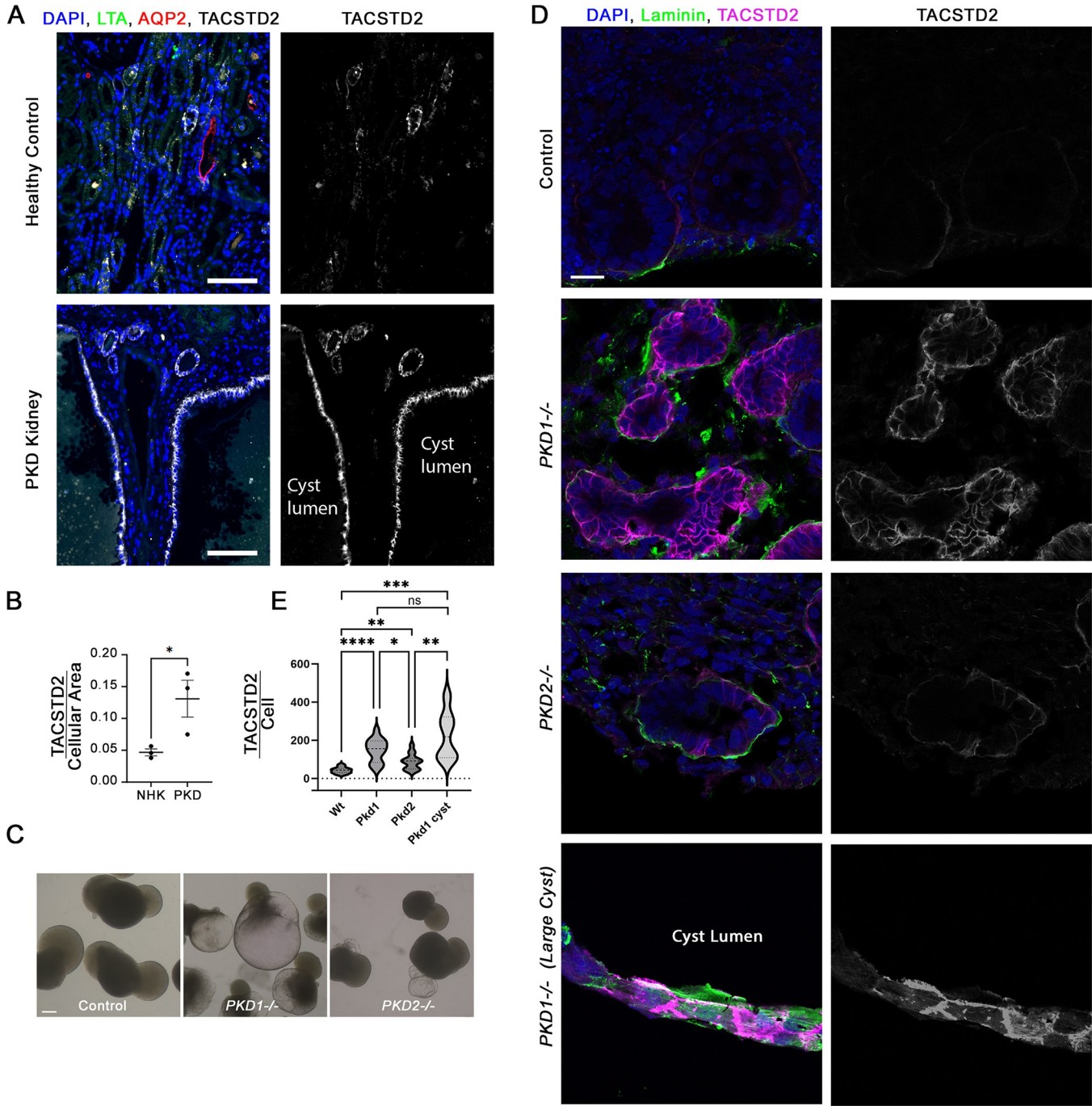

**Fig 5. Characterizing *TACSTD2* expression in human kidney epithelium.** (A) Healthy and ADPKD human kidney sections were probed for TACSTD2 (white), collecting duct marker Aquaporin-2 (AQP2, red) and proximal tubule marker Lotus tetragonolobus Agglutinin (LTA, green). Nuclei were marked by DAPI (blue). The Tacstd2 channel is shown in grey scale in the right image of each pair. Images captured by Zeiss AxioScan.Z1 with 20x objective. Scale bar is 100 microns. See S5 Fig. for images of individual channels. (B) Mean intensity of TACSTD2 fluorescence was measured for normal healthy kidney (NHK) and ADPKD human kidney samples normalized by cellular area. NHK n = 3; ADPKD n = 3. Unpaired t-test was performed. * p = 0.045. (C) Brightfield images of organoids derived from control, *PKD1*$^{-/-}$, and *PKD2*$^{-/-}$ mutant cells. Scale bar is 200 microns. (D) Immunofluorescence of organoid sections stained for TACSTD2 (fuchsia or white), laminin (green) and DAPI (blue). Scale bar is 20 microns. (E) Quantification of the images shown in D. Significance as compared to control *p<0.05; **p<0.01; ***p<0.001; ****p<0.0001; ns not significant by Welch and Brown-Forsythe ANOVA.

is deleted. In both Kunnen *et al.* [16] and Zhang *et al.* [17], the animals are mature at the time of deletion, while in our study and *Woo et al.'s* [15] the animals lose polycystin in the early postnatal stage when mouse kidneys are still developing. Humans with ADPKD mutations lack polycystin during kidney development *in utero*. It is possible that *Tacstd2/TACSTD2*'s role in cyst initiation may depend on developmental timing. The discrepancies between studies highlight the importance of using an array of genetic models, targets, and time points, to elucidate cyst initiators.

To characterize how Tacstd2 is regulated during early stage of cystogenesis we immunostained tissues for Tacstd2. Consistent with our RNAseq study, we found staining slightly elevated in experimental kidneys compared to controls at P6 and dramatically elevated at P10. Mouse findings were consistent with human kidney tissues and organoids, suggesting that TACSTD2 is a relevant target in human disease.

Tacstd2, also known as trophoblast cell surface antigen 2 (Trop2), is a transmembrane protein originally identified in trophoblast cells [28] and subsequently found to be highly expressed in many solid tumors [29]. Tacstd2 expression is generally low in most tissues but is seen in cells with stem cell properties [29]. It appears to be an oncogene as overexpression increases proliferation of cancer cells with low basal expression and knockdown with siRNA reduces growth of cancer cells with high expression [30]. Its ability to mark cancer cells has led to the development of antibody drug conjugate therapies to deliver cytotoxic payloads to cancer cells [10]. The FDA approved drug, Sacituzumab govitecan (Trodelvy), which is a conjugate to a topoisomerase I inhibitor, is in clinical use to treat metastatic triple-negative breast cancer and advanced or metastatic urothelial carcinoma. Side effects of nausea, fatigue, alopecia, febrile neutropenia, myelosuppression, and diarrhea leading to dehydration and acute kidney injury were observed but other kidney toxicities were not reported. A second drug Datopotamab Deruxtecan is being explored in several clinical trials and has shown reduced side effects. Kidney effects were not reported [31]. To the best of our knowledge Tacstd2 expression has not previously been explored in relation to ADPKD. While antibody drug conjugates to topoisomerase inhibitors may be too toxic for long term use in PKD patients, adapting the conjugates to carry anti-cyst payloads such as rapamycin could increase the efficacy of an anti-cyst drug while reducing its off-target effects.

Two mouse *Tacstd2* mutant lines have been generated. Contrary to expectations, the mice were viable and in the first report, no pathology was observed [32]. The second line generated by KOMP shows microphthalmia and small testis but no other phenotypes (https://www.mousephenotype.org/data/genes/MGI:1861606). Humans with pathological variants in *TACSTD2* develop gelatinous drop-like corneal dystrophy, which is a degenerative disease of the cornea [33]. In the cornea, TACSTD2 is thought to bind to claudins 1 and 7 to maintain the structure of the tight junctions. When TACSTD2 is defective, the cornea becomes more permeable and accumulates amyloid, ultimately leading to vision loss [34]. The relatively mild phenotypes of mouse and humans lacking TACSTD2, supports the idea that this gene product could be targeted without causing major off target effects.

In the context of PKD, it is interesting to note that antibodies binding the extracellular domain of Tacstd2 transiently increase cytoplasmic calcium concentration, possibly from an intracellular store [35]. The cytoplasmic tail of Tacstd2 contains a $PIP_2$ binding site and numerous pathways are affected by Tacstd2. In tumor cells, AKT signaling shows strong correlation with Tacstd2 expression [36] but MAPK/ERK, ErbB, TGFβ, Wnt/β-catenin, JAK/STAT, and integrin signaling have all been implicated as downstream targets of Tacstd2 [37]. Wnt/β-catenin signaling is particularly interesting as this pathway is disturbed in PKD. β-catenin binds directly to Tacstd2 and Tacstd2 over expression promoted nuclear accumulation of β-

catenin [38]. This observation could have implications for driving cyst formation in PKD as expression of activated β-catenin is sufficient to drive cyst formation [39].

In conclusion, our work demonstrates that Tacstd2 is elevated early in cystic kidney tissue as well as human polycystic kidneys. The insights gained from studies of breast and other aggressive cancers suggest that Tacstd2 is a promoter of cell proliferation and could play a key role in cyst formation. Further studies are needed to elucidate the mechanistic role of Tacstd2 in PKD. Our work suggests that Tacstd2 is a promising target in PKD and should be explored as a therapeutic target. More immediately, our work indicates that patients with PKD who receive anti-TACSTD2 therapy should be monitored for kidney effects.

## Materials and methods

### Ethics statement

Mouse studies were approved by the Institutional Animal Care and Use Committee of the University of Massachusetts Chan Medical School (PROTO201900265).

Human kidney tissue samples from nephrectomized kidneys were received from the Maryland PKD Research and Translational Core Center (part of the NIDDK U54 PKD-RRC). Human sample use was reviewed by the University of Maryland School of Medicine Institutional Review Board (IRB) and determined not to be human research, requiring no further IRB review. Thus, the Ethics Committee/IRB of the University School of Medicine waived ethical approval of this work.

### Mouse studies

Mouse strains used include $Pkd2^{flox}$ [11], $Cagg\text{-}Cre^{ER}$ [40], and $Rosa26\text{-}Cre^{ERT2}$ [41]. $Pkd2^{flox}$ was converted to $Pkd2^{null}$ by crossing to the germline $Stra8\text{-}Cre$ [42] to delete the floxed exons. All mice were C57BL/6J congenics maintained by backcrossing to C57BL/6J purchased from Jackson Laboratory (Bar Harbor Maine, USA). These studies were approved by the Institutional Animal Care and Use Committee of the University of Massachusetts Chan Medical School.

For RNA sequencing, pups segregating $Pkd2^{+/flox/null}$ and $Cagg\text{-}Cre^{ER}$ were dosed with tamoxifen (50 mg/kg) by intraperitoneal injection on postnatal day 2 (P2). On postnatal day 6 (P6) or day 10 (P10), paired kidneys were immediately frozen in RNA later, and total RNA was isolated as previously described [43] (Fig 1A).

For histology of P21 kidneys (Fig 4 only), mothers were dosed with tamoxifen (200 mg/kg) by oral gavage on P2, 3, and 4. The $Pkd2^{flox/flox}$, $Rosa26\text{-}Cre^{ERT}$ pups remained with nursing mothers until euthanasia at P21 as previously described [22].

### RNA sequencing

**mRNA.** Libraries were prepared for 100bp paired-end sequencing by BGI Genomics (Yantian District, Shenzhen, China). Read quality assessment, alignment and counting were performed using DolphinNext pipeline [44]. The modules used were FastQC v0.11.7 and kallisto v0.46.1 [45]. Differential expression analysis was performed using DESeq2 [46]. Very low expression genes were excluded using cutoff counts per million (CPM) <2 in at least 48 samples. Inclusion criteria for significant differentially expressed genes (DEGs) consisted of FDR < 0.05.

**Small RNA.** Small RNA libraries were prepared for 50 bp single-end sequencing by BGI Genomics (Yantian District, Shenzhen, China). The sequences were demultiplexed and underwent bar code and adapter sequence removal using the SOAPnuke software [47]. The steps for

demultiplexing, bar code removal and adapter removal were carried out by BGI. Read quality assessment, alignment and miRNA prediction and quantification/gene expression calculation as well as differential expression were performed using the Omiras pipeline [48] which uses Bowtie for read mapping [49], mirDeep2 for finding novel miRNA [50], and DESeq for normalizing the data and calculating differential expression [51]. Inclusion criteria for significant differentially expressed microRNAs (DEmirs) consisted of raw read counts > 0, FDR < 0.05, and identified as mature or stem-loop microRNAs. Our data was compared to previously published studies [52,53,54,55,56].

## Functional Enrichment Analysis

We performed functional enrichment analysis on ranked gene lists using g:Gost, a specialized module within the g:Profiler framework tailored for conducting gene set enrichment analysis based on user-defined query parameters [12]. All gene lists were ranked according to p adjusted values and queried using identical parameters outlined below. Gene lists were P6_All (91), P6_Unique (17), CIC_Common_Genes (74), P10 _All (top 1000 by padj rank), P10_Unique (top 1000 by padj rank). PKD2 was excluded from all input lists.

The analysis conducted by g:Gost was based on the following query parameters:
Version: e111_eg58_p18_30541362
Organism: Mus musculus (mmusculus)
All Results: Excluded (FALSE)
Ordered: TRUE
Exclude IEA Annotations: FALSE
Sources: Gene Ontology Molecular Function (GO:MF), Gene Ontology Cellular Component (GO:CC), Gene Ontology Biological Process (GO:BP), Kyoto Encyclopedia of Genes and Genomes (KEGG), Reactome (REAC), WikiPathways (WP)
Multiquery: Excluded (FALSE)
Numeric Namespace: ENTREZGENE_ACC
Domain Scope: Annotated
Measure Underrepresentation: Excluded (FALSE)
Significance Threshold Method: False Discovery Rate (FDR)
User Threshold: 0.01
No Evidences: Excluded (FALSE)
Highlight Results: Excluded (FALSE)

## Comparison to published bulk RNAseq data

Comparisons to published mouse datasets were performed by filtering publicly available data for genes with FDR <0.05 and comparing them against genes in our dataset with FDR <0.05. Simple linear regression analysis was performed using GraphPad Prism 9, Version 9.5.1. Datasets include:

- [16] S9 Table. Fold change of differentially expressed genes in mice deleted of *Pkd1* by KspCad-Cre[ERT2] compared to *Pkd1* wild type controls.

- [15] S1 Table. Differentially expressed genes in mice deleted of *Pkd1* by HoxB7-Cre.

- [15] S3 Table. Differentially expressed genes in mice deleted of *Pkd2* by HoxB7-Cre.

- [17] S2 Table. Differentially expressed genes in mice deleted of *Pkd2* by Pax8[rtTA],TetO-Cre.

## Comparison to published single-cell RNAseq analysis

**Gene Expression Omnibus (GEO) murine.** Our present work examines changes in whole-kidney gene expression following *Pkd2* deletion but does not identify which of the diverse kidney cell types express the gene. To determine whether there is any cell-specific localization of candidate genes, we reanalyzed data from Ransick et al. [21], who identified enriched transcriptome profiles within individual kidney cell types. Murine transcriptomes were obtained as 12 objects from GSE129798 [21]. Raw data was filtered to retain cells that expressed between 200 and 6,000 unique genes and a mitochondrial fraction less than 10%. 31056 cells from the cortex, outer medulla, and inner medulla fit these criteria. Each dataset was further pre-processed by log normalizing each dataset with a scale factor of 10,000 and finding 2,000 variable features using the default 'vst' method in seurat3.0 [57]. The dataset was then scaled to regress out differences driven by unequal total counts captured and mitochondrial capture. The top 40 principal components were used for construction of a Uniform Manifold Approximation and Projection (UMAP) and for neighbor finding. During clustering we utilized a resolution parameter of 0.5. The resulting 28 Louvain clusters were visualized in 2D and/or 3D space and were annotated using known biological cell-type markers [19].

**GEO wild-type and ADPKD human.** To determine if candidate cyst initiation genes from our mouse *Pkd2* model were also expressed in human kidneys, we reanalyzed published kidney transcriptome data from human PKD patients and from healthy controls. Data were obtained as 13 objects from GSE185948 [18].

The data was filtered to only retain cells that expressed between 500 and 6,000 unique genes and a mitochondrial fraction less than 10%. Heterotypic doublets were identified with Doubet-Finder v2 assuming 8% of cells represented heterotypic doublets. The datasets were then integrated in Seruat using "IntegrateData". These parameters resulted in retention of 56,161 cells from the healthy controls and 69,385 cells from PKD patients. Each dataset was further pre-processed by log normalizing each dataset with a scale factor of 10,000 and finding 2,000 variable features using the default 'vst' method in seurat3.0 [57]. The top 50 principal components were used for initial integration. The dataset was then scaled to regress out differences driven by unequal total counts captured and mitochondrial capture. The ADPKD and Healthy control datasets were then integrated with batch effect correction using Harmonyv1. The top 15 principal components were used for construction of a UMAP and for neighbor finding. During clustering we utilized a resolution parameter of 0.5. The resulting Louvain clusters were visualized in 2D and/or 3D space and were annotated using known biological cell-type markers [19].

For pseudobulk analysis cells were grouped by their disease state, and the Seurat function "AverageExpression" was used to average expression across all cell types present in each sample. This data was then rescaled and visualized in heatmaps. Differentially expressed genes between the ADPKD and healthy controls were calculated using "FindMarkers" filtering for transcripts with a log2FC greater than 0.25 and expression in greater than 20% of cells.

## Kidney organoid differentiation

Human induced pluripotent stem (IPS) cells were obtained from PKD Research Resource Consortium University of Maryland (Baltimore, Maryland) and verified to be of human origin and to have the correct genotype by sequencing of the regions of *PKD1* and *PKD2* targeted by the guides.

To produce organoids, on day 0, IPS cells were plated at 7,300 cells per $cm^2$ in mTeSR1 media (Stemcell 85850, Kent, WA) on vitronectin (ThermoFisher Scientific A14700, Carlsbad CA) coated 6-well plates. On day 1, differentiation was initiated by replacing medium with

STEMdiff APEL2 Medium (Stemcell 05275, ent, WA) containing 8μM CHIR99021 (Abcam ab120890, Cambridge, MA) for four days. The media was replaced with fresh media every two days. On day 5, cells were washed with Gibco DPBS, no calcium, no magnesium (Thermo-Fisher Scientific), and media replaced with fresh APEL2 media containing 50ng/mL of recombinant human FGF9/GAF protein (Abcam ab50034) and 1μg/ml of 0.2% heparin sodium salt in dissolved in PBS (Stemcell 07980, Kent, WA) for three days. On day 7, the media was replaced with APEL2 containing 5μM CHIR99021 for 1 hr. Cells were then washed and dislodged using TrypLE Express without phenol red (Thermo-Fisher 12604013) and then spun down. The cells were then resuspended in APEL2 containing FGF9 50ng/mL and Heparin 1μg/mL, transferred to an Ultra-Low Attachment Multiple Well Plate (Costar size 6 well, flat bottom CLS3471, Allentown, PA) and placed on an orbital shaker at 100 rpm in 37°C 5% $CO_2$ incubator. Media was changed every two days. After five days, the medium was replaced with APEL2 without additions and changed every two days. Organoids were typically harvested 14 days later. This protocol is a modification of Ruiter *et al.* [23].

## Histology

**Mouse tissues.**  Tissues were fixed by immersion overnight in 10% formalin (Electron Microscopy Sciences) in phosphate-buffered saline and then embedded in paraffin. Sections were deparaffinized and stained with hematoxylin and eosin (H&E).

Images of H&E-stained sections were obtained with a Zeiss Axio Scan.Z1 slide scanner with brightfield capabilities using the 20X objective.

For immunofluorescent staining, sections were deparaffinized, antigens were retrieved by autoclaving for 30 min in 10 mM sodium citrate, pH 6.0 and stained with primary antibodies diluted in TBST (10 mM Tris, pH 7.5, 167 mM NaCl, and 0.05% Tween 20) plus 0.1% cold water fish skin gelatin (Sigma-Aldrich). Alexa Fluor–labeled secondary antibodies (Invitrogen) were used to detect the primary antibodies. Primary antibodies used included Aquaporin-1 (1:100, Santa Cruz, sc-32737), Aquaporin-2 (1:200; Sigma, SAB5200110), Tacstd2/Trop-2 (F-5) (1:250, Santa Cruz, sc-376181). FITC-conjugated lectins were added with secondary antibodies: Lotus tetragonolobus agglutinin (LTA, 1:50, Vector Labs) or Dolichos biflorus agglutinin (DBA, 1:20, Vector Labs). Nuclei were labeled with 4′,6-diamidino-2-phenylindole (DAPI).

Fluorescent images were obtained with a Zeiss LSM900+ Airyscan microscope. Fluorescent slide scans were obtained using Zeiss Axio Scan.Z1 slide scanner at 20X objective.

**Human tissues.**  Paraffin slides of kidney sections from 3 male ADPKD samples and 3 healthy male controls were obtained from the Polycystic Kidney Disease Research Resource Consortium at the University of Maryland (Baltimore, Maryland). No genotype or additional information is available for these specimens. Slides were deparaffinized, antigens retrieved, and stained as described above using Aquaporin-2 (1:200; Sigma, SAB5200110), Tacstd2/Trop-2 (F-5) (1:250, Santa Cruz, sc-376181), FITC-conjugated Lotus tetragonolobus agglutinin (LTA, 1:50, Vector Labs).

Fluorescent images were obtained with a Zeiss LSM900+ Airyscan microscope. Fluorescent slide scans were obtained using Zeiss Axio Scan.Z1 slide scanner at 20X objective.

**Organoids.**  Plates were tilted until kidney organoids settled to the bottom and medium was slowly removed with a 5ml pipette. Organoids were washed with 2 ml of PBS and again allowed to settle to the bottom of the dish. After removal of the PBS, 2ml of 4% paraformaldehyde in PBS was added for 1 hr. Following this, paraformaldehyde was removed and organoids were resuspended in 500 μL of PBS in a microcentrifuge tube. The PBS was replaced with 200 μL of 30% sucrose in 100mM Sorensen's Phosphate Buffer. After 24 hrs, the organoids

were transferred to a cryomold and embedded in Tissue Plus O.C.T. Compound (Fisher HealthCare 4585). Embedded organoids were frozen on dry ice. Cryoblocks were sectioned at 10 microns and collected on Superfrost Plus Microscope slides (Fisherbrand, 12-550-150).

Slides were washed with PBS containing 0.1% Tween20 (PBS-T) at 37˚C for 10 min. The organoids fragments on the slide were circled with SuperHT Pap Pen (Biotium 22006, Fremont, CA). 300 µL of blocking solution (5% donkey serum in PBS-T) was applied to each slide and incubated at room temperature in a humidified chamber for 1 hr. After removal of the blocking solution, primary antibodies diluted in 5% BSA in PBS-T were incubated overnight at room temperature in a humidified chamber. After three washes, secondary antibodies diluted in 5% BSA in PBS-T were added for 2 hrs in the dark. Then slides were washed three times with PBS-T for 30 min. Slides were air-dried for 5 min and coverslips were mounted with Prolong Gold Antifade with DAPI (ThermoFisher Scientific, P36935).

Fluorescent images were obtained using a ZEISS LSM900+ Airyscan microscope and analyzed by ZEN lite software.

## Supporting information

**S1 Table. Differentially expressed genes at P6 (Excel Spreadsheet).**
(XLSX)

**S2 Table. Differentially expressed genes at P10 (Excel Spreadsheet).**
(XLSX)

**S3 Table. Overlap of differentially expressed genes at P10 and human ADPKD (Excel Spreadsheet).**
(XLSX)

**S4 Table. Functional enrichment of genes at P6 and P10.**
(XLSX)

**S5 Table. Functional enrichment of genes at P6 and P10.**
(XLSX)

**S6 Table. Functional enrichment of genes at P6 and P10.**
(XLSX)

**S7 Table. Functional enrichment of genes at P6 and P10.**
(XLSX)

**S8 Table. Functional enrichment of genes at P6 and P10.**
(XLSX)

**S1 Fig. Small RNAseq of *Pkd2* experimental kidneys vs control at P10 identifies micro-RNAs with known association in PKD.** (A) Volcano plot showing P10 differentially expressed microRNAs (Omiras). Colored dots indicate FDR <0.05, red are upregulated and blue are downregulated in Experimental vs. Control. Plot was cropped for better visualization of data. (B) Table indicates microRNAs described in (A) that were previously cited in literature with relation to polycystic kidney disease. Citation refers to the individual microRNA or its related family or cluster. Red are upregulated and blue are downregulated in our results.
(PDF)

**S2 Fig. Differentially expressed mRNAs in pre-cystic and early cystic mouse kidneys.** (A) Functional enrichment analysis for P6 unique DEGs, P10 unique DEGs, and those common to both lists (CICs). GO, KEGG, Reactome, and WikiPathways databases were probed using g:

profiler. Top functions are shown and listed, ranked by p adjusted values. (B) Heatmap comparing differential gene expression for cyst initiation candidates (CICs) in our data (P6, P10) and published pre-cystic mouse data [WP_Pkd2_P7/3/1, WP_Pkd1_P7/3/1 [15], ZS [17], KP2/3/6 [16]. *Tacstd2* identified with label and black box. (C) Barplots showing the number of differentially expressed genes (False discovery rate < 0.05) identified in data from the described studies. Our study: P6 and P10 following Pkd2 deletion. Woo *et al*. [15]: P1, P3, and P7 following Pkd1 or Pkd2 deletion. Kunnen *et al*. [16]: Weeks 2, 3, and 6 following Pkd1 deletion. Pkd1 models represented by blue bars). Pkd2 models represented by black bars. (PDF)

**S3 Fig. *Tacstd2* expression in cystic epithelium.** (A) Dotplot of marker genes supporting cell type assignments in Fig 3E. Markers according to Chen et al. [19]. Dot radius represents percent of cells in cluster expressing the gene. Opacity scale indicates average gene expression level within expressing cells. Row normalized. (B) UMAP from Fig 3E with cell type assignments as labeled by Ransick et al. [21]. (C) Dotplot of marker genes used to annotate S3B Fig. Dot radius represents percent of cells in cluster expressing the gene. Opacity scale indicates average gene expression level within expressing cells. Row normalized. (D) DotPlot visualizing the localization of CICs to the murine kidney cell atlas. Dot radius represents percent of cells in cluster expressing the gene. Opacity scale indicates average gene expression level within expressing cells. Row normalized. Main bars (far left) represent increased expression (red) or decreased expression (blue) in our data; sub bars (left) represent epithelial restricted expression (magenta), broad expression (green), and stromal/indeterminate (gray). Black arrow points to *Tacstd2*, an epithelial cyst initiating candidate. Blue arrow points to PKD2. (PDF)

**S4 Fig. Tacstd2 expression in cystic mouse renal epithelium.** (A) All scans used for quantification described in Fig 4. Scale bars 1mm. (PDF)

**S5 Fig. Characterizing *TACSTD2* expression in human kidney epithelium.** Individual channels of image in Fig 5A. Healthy and ADPKD human kidney sections were probed for TACSTD2 (white), collecting duct marker Aquaporin-2 (AQP2, red) and proximal tubule marker Lotus tetragonolobus Agglutinin (LTA, green). Nuclei were marked by DAPI (blue). Images captured by Zeiss AxioScan.Z1 with 20x objective. Scale bars are 100 microns. (PDF)

## Acknowledgments

We thank Dr. Owen M. Woodward from the Polycystic Kidney Disease Research Resource Consortium at the University of Maryland for sections of human kidney.

## Author Contributions

**Conceptualization:** Abigail O. Smith, William Tyler Frantz, Kenley M. Preval, Yvonne J. K. Edwards, Julie A. Jonassen, Gregory J. Pazour.

**Data curation:** Abigail O. Smith, William Tyler Frantz, Kenley M. Preval, Yvonne J. K. Edwards.

**Formal analysis:** Abigail O. Smith, William Tyler Frantz, Kenley M. Preval, Yvonne J. K. Edwards, Julie A. Jonassen, Gregory J. Pazour.

**Investigation:** Abigail O. Smith, William Tyler Frantz, Kenley M. Preval, Yvonne J. K. Edwards, Julie A. Jonassen, Gregory J. Pazour.

**Supervision:** Craig J. Ceol.

**Visualization:** Abigail O. Smith.

**Writing – original draft:** Abigail O. Smith, William Tyler Frantz, Kenley M. Preval, Gregory J. Pazour.

**Writing – review & editing:** Abigail O. Smith, William Tyler Frantz, Kenley M. Preval, Yvonne J. K. Edwards, Craig J. Ceol, Julie A. Jonassen, Gregory J. Pazour.

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
