## [Decision Letter · Decision Letter 0]

29 Jul 2024

Dear Greg,

Thank you very much for submitting your Research Article entitled 'The Tumor-Associated Calcium Signal Transducer 2 (TACSTD2) oncogene is upregulated in pre-cystic epithelial cells revealing a new target for polycystic kidney disease' to PLOS Genetics.

The manuscript was fully evaluated at the editorial level and by independent peer reviewers. The reviewers appreciated the attention to an important problem, but raised some substantial concerns about the current manuscript. All reviewers agree about the novelty and significance of this work: TACSTD2 is a novel ADPKD marker that is a potential therapeutic target to prevent cyst formation. There is some disagreement about experimental support and mechanistic understanding for TACSTD2 as a cyst-initiation factor. Authors should address Reviewer 1s specific comments regarding the site of TACSTD2 expression. To address Reviewer 2s serious concerns about lack of experimental support and mechanistic understand, Reviewer 3 points out that it is important to clearly distinguish "coincidence"/"correlation" from "causality". Authors may want to include a limitation of study/future directions discussion section to address the concerns of Reviewer 2. Based on the reviews, we will not be able to accept this version of the manuscript, but we would be willing to review a much-revised version. We cannot, of course, promise publication at that time.

If you decide to revise the manuscript for further consideration at PLOS Genetics, please aim to resubmit within the next 60 days, unless it will take extra time to address the concerns of the reviewers, in which case we would appreciate an expected resubmission date by email to plosgenetics@plos.org.

To resubmit, log into your Editorial Manager account and select the option 'Revise Submission' in the 'Submissions Needing Revision' folder.

We are sorry that we cannot be more positive about your manuscript at this stage. Please do not hesitate to contact us if you have any concerns or questions.

Yours sincerely,

Maureen M. Barr

Academic Editor

PLOS Genetics

Giovanni Bosco

Section Editor

PLOS Genetics

Reviewer's Responses to Questions

**Comments to the Authors:**

Reviewer #1: To find renal cyst initiators, Smith et al. perform transcriptomics on pre-cystic and early-to-moderately cystic kidneys of juvenile Pkd2 conditional ko mice. By comparing with other published ADPKD transcriptomes, including human ADPKD, they identify TACSTD2 as a potential novel drug candidate. They show increased expression of TACSTD2 in tubular epithelia of pre-cystic, moderately and severely mouse cystic kidneys, as well as in cyst-lining cells of human ADPKD kidneys. Using an iPSC-derived organoid model, they also show that loss of PKD1 increases TACSTD2 expression. The manuscript is nicely written and identifies TACSTD2 as a novel ADPKD marker that is aberrantly expressed at early stages of cystogenesis, suggesting this may be a beneficial target to prevent cyst formation.

Specific comments:

1. In Fig 4A at P21, whole kidney images appear to show TACSTD2 expression lining cysts that are mostly at the outer edge of the kidney, while cells lining large cysts in the medulla appear to not express TACSTD2. In Fig 4C, please include images of inner cortex as well as of medulla and/or confirm that TACSTD2 is only cortical. If only cortical, how do authors envision TACSTD2 plays a role (or not) in medullary cysts?

2. Fig 4C. The human scRNAseq data show TACSTD2 is also expressed in loop of Henle (Fig 3G). To more fully characterize TACSTD2 expression, this reviewer recommends adding a loop of Henle marker in Fig 4C.

3. Fig 5A. Similarly, this reviewer recommends including a loop of Henle marker on the human ADPKD sections.

4. In abstract, please delete final sentence, which is speculative, and because association between ADPKD and cancer incidence is controversial (PMID: 24854270; PMID: 37542051). Alternatively, sentence or line of thought can be included in discussion.

5. P.3, line 73. Italicize genes: PKD1/Pkd1 and PKD2/Pkd2

6. P. 8 line 165, describe the Cre used and timepoint at which deletion of Pkd1/Pkd2 were induced in Kunnen et al. [16] and Zhang et al. [17], similar to how Woo et al. was described.

7. P. 8, line 179, “human data samples late stage ADPKD”. Please fix phrasing.

8. P. 9, line 185, change “We” to “we”

9. P. 12, lines 248-250. Two sentences sound a little contradictory. Recommend adding, “Yet,” or “However,” to beginning of 2nd sentence or rephrasing.

Reviewer #2: In this study, the authors investigate the role of Tacstd2 in the context of Autosomal Dominant Polycystic Kidney Disease (ADPKD) using an inducible global Pkd2 knockout mouse model. They identify Tacstd2 as significantly upregulated in renal epithelial cells of both mouse PKD kidneys and human PKD tissue biopsies and organoids, but low in healthy renal cells, proposing it as a potential target for PKD treatment. While the study provides valuable insights into expression dynamics of Tacstd2, an interesting oncogene and a novel player in the context of ADPKD, the study is mostly a descriptive study with limited mechanistic evidence supporting the statement that Tacstd2 is indeed a cyst initiating factor. Importantly, no functional experiments were designed to confirm Tacstd2 is required for cystogenesis and that Taxstd2 deficiency ameliorate cystogenesis. Specifically:

1. The use of a global-inducible system (Cagg-CreER) for Pkd2 deletion is considered as a weakness. A segment-specific inducible Cre/ERT2, KspCad or Aqp2 system should be used to minimize noise introduced by ubiquitous Pkd2 deletion across the kidney, potentially may revealing more candidates in pathogenic renal tubules.

2. The physiology of P6 kidney is very different than adult kidneys in which ADPKD occurs. An adult-inducible PKD rodent model or hylomorphic Pkd1RC/RC line would be better for identifying cyst-initiating factors.

3. In P6 kidneys, the study identifies a limited number of differentially expressed genes (DEGs) with subtle fold changes, including Tacstd2 which shows only a modest increase (~25% more than controls). This is consistent with the IF images at P6 (Fig. 4A) that Tacstd2 indeed expresses in healthy kidneys. This challenges the assertion that Tacstd2 acts as a primary cyst-initiating factor.

4. In vitro and in vivo functional characterizations are needed to validate Tacstd2 as a critical cyst initiating factor, which is central statement of the current study. These experiments would provide mechanistic insights into Tacstd2's role in cystogenesis and clarify whether its deficiency ameliorates cyst formation.

5. While the cross-referencing with human ADPKD data in Figure 3 is informative, it should be noted that this data originates from cystic kidneys rather than pre-cystic stages. Thus, validating Tacstd2 as a cyst-initiated factor but not a secondary target during cystogenesis is important.

Reviewer #3: In their study "The Tumor-Associated Calcium Signal Transducer 2 (TACSTD2) oncogene is upregulated in pre-cystic epithelial cells revealing a new target for polycystic kidney disease", Smith AO et al. describe an mRNA sequencing-based approach around the timepoint of initial cystogenesis in an inducible Pkd2 knockout mouse model that leads to the identification of Tacstd2.

The authors present a well-conceived and technically well performed investigation addressing a large unmet need in ADPKD therapeutics development; namely the understanding of distinguishing molecular features of early cystic epithelia. The authors should be applauded for this nice study, and it should be considered for publication in PLOS Genetics after minor revision.

The mouse model is well suited to address the experimental questions and the RNA collection, sequencing and data processing and analysis, as well as the comparative analysis with other published data sets is state of the art. The authors then validate the upregulation and specificity of TACSTD2 in cystic epithelia in human ADPKD tissue and in an organic model. I do not have any comments regarding the well conducted technical aspects of the study.

The authors need to be careful about the right language to clearly distinguish "coincidence"/"correlation" from "causality". At various point in the manuscript (for example, p.6 ln.123 "factors that PROMOTE cyst initiation"; p.7 ln.156 "CICs most likely to DRIVE early cystic transformation") there is the risk of overstatement (the authors correct it later in ln.126 "cyst initiation CANDIDATES). The paragraph p.9 ln.200 "Validating Tacstd2 as a cyst initiation candidate" is misleading, because the reader may get the impression that a causative mechanistic role if Tacstd2 in cyst initiation is being validated. Instead, the authors only demonstrate specificity of Tacstd2 as a marker in early cystic epithelia.

It seems important to convey the message that Tacstd2 (and all other DEGs) coincide in their individual expression pattern with the time point of cystogenesis, but that causality can only be ascribed if further experiments prove it necessary for early cyst formation. Obviously, for therapeutic targeting of early cystic epithelia by ADCs, identifying differentially expressed specific surface markers is fully sufficient, regardless of their causative contribution to the cystogenic process.

Please read your own work carefully under that aspect and re-write those passages where the distinction of causality and correlation is not clear.

**Have all data underlying the figures and results presented in the manuscript been provided?**

Reviewer #1: Yes

Reviewer #2: Yes

Reviewer #3: Yes

PLOS authors have the option to publish the peer review history of their article (what does this mean?). If published, this will include your full peer review and any attached files.

Reviewer #1: No

Reviewer #2: No

Reviewer #3: No

---

## [Decision Letter · Decision Letter 1]

30 Oct 2024

PGENETICS-D-24-00597R1The Tumor-Associated Calcium Signal Transducer 2 (TACSTD2) oncogene is upregulated in pre-cystic epithelial cells revealing a new target for polycystic kidney diseasePLOS Genetics Dear Dr. Pazour, Thank you for submitting your manuscript to PLOS Genetics. After careful consideration, we feel that it has merit but does not fully meet PLOS Genetics's publication criteria as it currently stands. Therefore, we invite you to submit a revised version of the manuscript that addresses the points raised during the review process. Your revised manuscript was evaluated by two reviewers and the editor. To address Reviewer 2s lingering concern that TACSTD2 is not a cyst-initiating factor, please implement Reviewer 2s points 2 and 3 in the text. These editorial changes, combined with the addition of the limitation of studies section and use of the term "cyst initiating candidates" in this revised manuscript, are sufficient. You should include individual channels in Figure 5, per Reviewer 2. Please submit your revised manuscript within 30 days Nov 29 2024 11:59PM. If you will need more time than this to complete your revisions, please reply to this message or contact the journal office at plosgenetics@plos.org. Please include the following items when submitting your revised manuscript:*
A rebuttal letter that responds to each point raised by the editor and reviewer(s). You should upload this letter as a separate file labeled 'Response to Reviewers'. This file does not need to include responses to formatting updates and technical items listed in the 'Journal Requirements' section below.*
A marked-up copy of your manuscript that highlights changes made to the original version. You should upload this as a separate file labeled 'Revised Manuscript with Track Changes'.*
An unmarked version of your revised paper without tracked changes. You should upload this as a separate file labeled 'Manuscript'. If you would like to make changes to your financial disclosure, competing interests statement, or data availability statement, please make these updates within the submission form at the time of resubmission. Guidelines for resubmitting your figure files are available below the reviewer comments at the end of this letter. We look forward to receiving your revised manuscript. Kind regards, Maureen M. BarrAcademic EditorPLOS Genetics Giovanni BoscoSection EditorPLOS Genetics Aimée DudleyEditor-in-ChiefPLOS Genetics Anne GorielyEditor-in-ChiefPLOS Genetics**Reviewers' comments:** Reviewer's Responses to Questions

**Comments to the Authors:**

Reviewer #1: The authors have addressed all concerns.

Reviewer #2: 1. In light of my previous critiques regarding whether TACSTD2 can be classified as a CIC, we should agree that the P6 data is crucial, as P10 kidmeys already show microscopic cysts, especially in renal cortex. However, the observation in P220 that “At P6, the 220 differences did not reach significance (Fig 4C)” renders the conclusion that TACSTD2 is upregulated in pre-cystic stage and potential a CIC invalid.

2. Consequently, the authors should reconsider the title “…upregulated in pre-cystic kidneys…” and the abstract statement “Tumor-associated calcium signal transducer 2 (Tacstd2) stood out as an epithelial-expressed gene with elevated levels prior to cystic transformation that further increased with disease progression”. This is a promising candidate for PKD, but at this stage, without functional characterization, TACSTD2 should primarily be described as a potential key player but not initiator in the early stages of cystogenesis.

3. In line 280, I suggest changing “To verify that Tacstd2 is a bona fide epithelial cyst initiating candidate…” to “To characterize how Tacstd2 is regulated during early stage of cystogenesis,” since none of the experiments conducted confirm that Tacstd2 is a bona fide CIC.

4. Multichannel staining of human patient kidneys in Fig 5 should include individual channels to enhance readers' understanding of TACSTD2 in different renal tubular segments in both healthy and cystic kidneys.

**Have all data underlying the figures and results presented in the manuscript been provided?**

Reviewer #1: Yes

Reviewer #2: Yes

PLOS authors have the option to publish the peer review history of their article (what does this mean?). If published, this will include your full peer review and any attached files.

Reviewer #1: No

Reviewer #2: No

 **Figure resubmission:** While revising your submission, please upload your figure files to the Preflight Analysis and Conversion Engine (PACE) digital diagnostic tool, https://pacev2.apexcovantage.com/. PACE helps ensure that figures meet PLOS requirements. To use PACE, you must first register as a user. Registration is free. Then, login and navigate to the UPLOAD tab, where you will find detailed instructions on how to use the tool. If you encounter any issues or have any questions when using PACE, please email PLOS at figures@plos.org. Please note that Supporting Information files do not need this step. If there are other versions of figure files still present in your submission file inventory at resubmission, please replace them with the PACE-processed versions. **Reproducibility:** To enhance the reproducibility of your results, we recommend that authors deposit laboratory protocols in protocols.io, where a protocol can be assigned its own identifier (DOI) such that it can be cited independently in the future. Additionally, PLOS ONE offers an option to publish peer-reviewed clinical study protocols. Read more information on sharing protocols at https://plos.org/protocols?utm_medium=editorial-email&utm_source=authorletters&utm_campaign=protocols

---

## [Editor Report · Decision Letter 2]

23 Nov 2024

Dear Dr Pazour,

We are pleased to inform you that your manuscript entitled "The Tumor-Associated Calcium Signal Transducer 2 (TACSTD2) oncogene is upregulated in cystic epithelial cells revealing a potential new target for polycystic kidney disease" has been editorially accepted for publication in PLOS Genetics. Congratulations!

Yours sincerely,

Maureen M. Barr

Academic Editor

PLOS Genetics

Giovanni Bosco

Section Editor

PLOS Genetics

Aimée Dudley

Editor-in-Chief

PLOS Genetics

Anne Goriely

Editor-in-Chief

PLOS Genetics

Comments from the reviewers (if applicable):

**Data Deposition**

http://datadryad.org/submit?journalID=pgenetics&manu=PGENETICS-D-24-00597R2

**Press Queries**

---

## [Editor Report · Acceptance letter]

28 Nov 2024

PGENETICS-D-24-00597R2 

The Tumor-Associated Calcium Signal Transducer 2 (TACSTD2) oncogene is upregulated in cystic epithelial cells revealing a potential new target for polycystic kidney disease 

Dear Dr Pazour, 

We are pleased to inform you that your manuscript entitled "The Tumor-Associated Calcium Signal Transducer 2 (TACSTD2) oncogene is upregulated in cystic epithelial cells revealing a potential new target for polycystic kidney disease" has been formally accepted for publication in PLOS Genetics! Your manuscript is now with our production department and you will be notified of the publication date in due course.

With kind regards,

Anita Estes

PLOS Genetics

On behalf of:
